# Numerical study of probe parameters on performance of a transonic axial compressor

**Asad Islam** [ID]*, **Hongwei Ma**

School of Energy and Power Engineering, Beihang University, Beijing, China

* asadislam@buaa.edu.cn

**Data Availability Statement:** All relevant data are within the manuscript.

**Funding:** The authors gratefully acknowledge the funds from the National Natural Science Foundation of China (Grant No. 51776011), and the National Science and Technology Major Project (2017-V-0016-0068).

## Abstract

The paper shows the effect of the probe on the performance of a transonic axial speed compressor. The unobstructed flow case with the experimental data was validated and used as a guide for all subsequent study cases. The aerodynamic performance for different probe parameters were calculated numerically using ANSYS-CFX. This covered the results on compressor output from changing probe axial positions, the radial immersion depths, the size of the probe, and the total number of probes. The findings were evaluated in relation to the total pressure ratio, performance, margin of deflation and stability. The velocity part distributions further showed that the probe block and raises the flow Mach value, which is the explanation why the compressor rotor's total pressure ratio is lost. In fact, the parameters of the sample will significantly influence the calculation outcomes and affect the standard margin. The range of stability was also affected, which changes the performance trend from the choke to the stall. Consequently, the collection of correct probe parameters with fewer impact on compressor output is addressed.

## 1 Introduction

The pneumatic probes find wide applications in the aerospace industry, because of their low cost, easy implementation and its capability to give reliable and stable measurements. However, the presence of probe enhances the flow-complexity when it is installed at upstream of the cascade that may affect the stability and engine performance [1, 2].

Prof. Wyler [3] studied the isolated probe blockage effects on the free-stream Mach number and pressure in open jets and closed tunnel. The results found were approximately the same for probe in both the free jets and closed tunnel. But the major difference lied in a closed tunnel, the probe decreases the pressure and increases the Mach number. Thus in order to get more accurate measurement results by the probes, the flow-blocking effect must be minimized. Reducing the blocking effect means reducing the size of the probe which demands for greater material strength and its design. For high Mach numbers, higher material strength is required to resist the more substantial airflow pressure and reduce the measurement deviations. Wang Hongwei [4] studied the blocking effect due to cylindrical probes in free jets and wind tunnels. The results indicated that for a blocking ratio of more than 0.06, a 1-deg error was generated for static-pressure angle, while there exists a critical Mach number in a high subsonic wind tunnel. Besides, the more severe blockages were a result of higher Mach number involving

**Competing interests:** The authors have declared that no competing interests exist.

shock wave and flow separations made the calibration impossible. The effects of different head geometries [5] on the angular range of a three-hole probe depicted that the cobra type probe had a wide range of measurement angles out of the three structures tested. However, the cylindrical type head extended the range with hole separation angle of 30 degrees. In addition to these studies, the numerical simulations have also been used to study the unsteady wake effects due to rotating struts [6] on turbine blades. The rotating probe [7] has different effects on the channel with a noticeable reduction in axial velocity and static pressure. It also had a negative impact on compressor performance along with the reduction in tip-vortex intensity [8].

The effects on low-speed axial compressor performance have largely been studied in the past decade. Our research group at Beihang University (BUAA) has also reported the experimental and numerical studies of probe effects on low-speed axial flow compressor stall characteristics [9] and downstream flow-field distortions [8] by placing the probe at half chord upstream with half blade span intrusion. The results indicated that the probe presence could vary the flow-structure and can lower the compressor stall margin. The numerical simulations of probe indicated that the main contributor in the measurement error lies in the blockage [10]. Later the numerical simulation study of airfoil probes [11] depicted the vortex formations along both sides of the blade. Nevertheless, the intensity of these vortices gradually decreased in the downstream, with the pressure side as the most affected.

Furthermore, from the industrial application point of view, the thorough knowledge of the flow field measurement is also essential. For example, after rapid expansion through the nozzle the moist air could condense. The difference of time between inertial recovery and temperature restoration is important where the velocity gradient is high, leading to a non-equilibrium condensation thermodynamics [12, 13]. The liquid mass fraction, which could impact the measuring accuracy, has been stated to be predominantly determined by non-equilibrium condensations [14, 15]. In addition, greater the degree of expansion, the greater the non-equilibrium condensate and a greater lack of irreversibility [16–19], as well as loss due to friction could affect the end results. The precise flow field prediction is therefore necessary to interpret the test results.

Summarizing the associated researches on the upstream probe and compressor performance effects, most scientists focused on the flow field of low-speed axial compressors. The transonic speed compressor has not yet been studied for the results of probe size variations, axial locations, probe immersion depths and the full annulus effects of the total number of probes. These parameters can largely influence the performance trend and can, due to their improper choice, become a source of measurement deviation. The unavailability of such research to concentrate on compressor performance, stability and stall margin is a novel development in current research. The new research presented in the current article therefore focuses on the impacts on aerodynamic compressor performance in terms of the total pressure ratio, adiabatic efficiency, stability and stable margin.

## 2 Computational setup

The performance effects on a transonic speed axial compressor Rotor–37 is calculated in the present study. The design total pressure ratio of the rotor is 2.106 at a mass flow rate of 20.19 *kg/s* while rotating at a design speed of 17.1887 *krpm* [20]. The total number of blades are 36 and the $k - \epsilon$ turbulence model [21–24] was used to solve the 3D steady compressible Navier-Strokes equation using a finite volume approach. Total pressure and total temperature values of 1 *atm* and 288*K* respectively were specified at the inlet (Station 1), whereas, the static pressure value at the outlet (Station 4) was varied to obtain the performance characteristics from

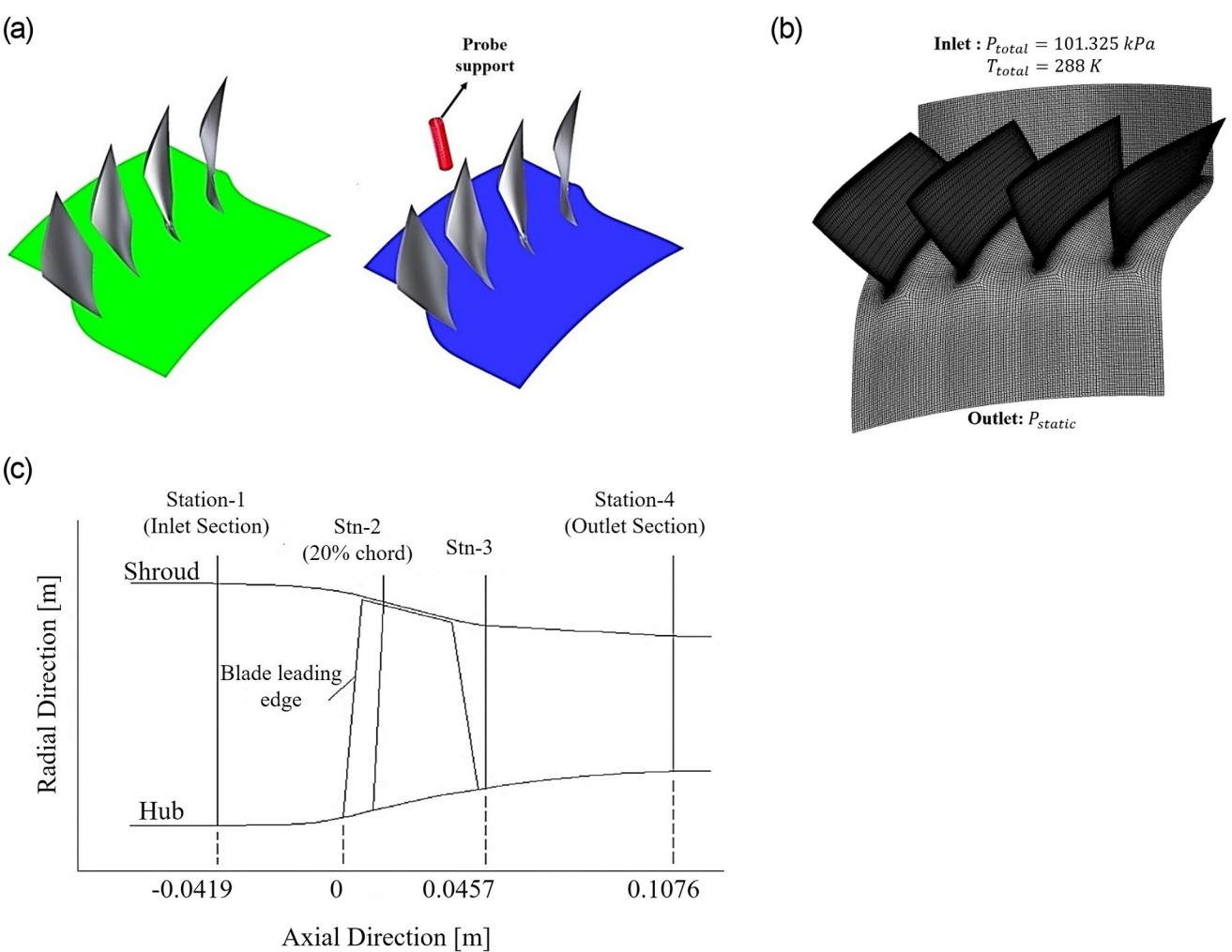

**Fig 1.** CFD model of (a) Clean flow (left) with the probe (right), (b) boundary conditions, and (c) measurement stations [28].

choke to stall inception, Fig 1. The governing equations of continuity, momentum and energy equations are as follows:

Continuity:

$$\frac{\partial \rho}{\partial t} + \nabla \cdot (\rho U) = 0 \tag{1}$$

Momentum:

$$\frac{\partial (\rho U)}{\partial t} + \nabla \cdot (\rho U \times U) = -\nabla p + \nabla \cdot \tau + S_M \tag{2}$$

Where, the stress tensor $\tau$ is related as;

$$\tau = \mu \left( \nabla U + (\nabla U)^T - \frac{2}{3} \delta \nabla \cdot U \right) \tag{3}$$

And the total energy equation:

$$\frac{\partial(\rho h_t)}{\partial t} - \frac{\partial \rho}{\partial t} + \nabla \cdot (\rho U h_t) = \nabla \cdot (\lambda \nabla T) + \nabla \cdot (U \cdot \tau) + U \cdot S_M + S_E \quad (4)$$

Where,

$$h_t = h + \frac{1}{2} U^2 \quad (5)$$

The viscous work term is represented using the term $\nabla \cdot (U \cdot \tau)$, while $U \cdot S_M$ is the term that represents work due to external momentum sources.

The first step considered the validation of rotor performance without the probe that is hereby referred to as the clean flow. It served as a basis for comparison of subsequent analysis of performance characteristics in the presence of probe. The influence of probe parameters on compressor performance is studied concerning variations in probe axial position, radial immersion height, probe size, and total probe number. All the subsections consider probe at the half circumferential position as the flow characteristics are not much influenced at this position as reported [25]. Furthermore, the first three subsections considered the total blades to total probe ratio of 4, except the last, where blade passages per probe were changed accordingly. Besides, the probe size is 8*mm* except for the third subsection, where the different probe sizes were studied and were changed accordingly. All the analyses were performed at the rotational design speed. The further details of the subsections are as follows:

1. Firstly the probe is located streamwise at different axial positions ranging from 20% (the closest) to 70% (the farthest location) while keeping the half intrusive probe height (50% of blade span).

2. Later, the axial probe position (half chord) is fixed, and variations due to intrusive heights (25% -full immersion) were studied.

3. Probe size is one of the important parameters in the study that was varied from 4*mm* to 14*mm* while keeping the same axial and radial probe position (*i.e.*, 50% axial chord and half immersion respectively).

4. Finally, the most critical parameter that is often neglected is the probe number. Six different blades-to-probe ratios were studied with respective blade passages per probe and the results were recorded while maintaining a fixed axial and span-wise height of the probe (*i.e.*, 50% chord and 50% span of the rotor blade respectively).

## 3 Results and discussions

The results are divided into four sections. Firstly, the clean flow case (*i.e.*, without the probe) is validated with the available experimental data. Then, the rotor characteristics curves were generated and compared with the clean flow reference case to calculate the relative influence. The stall margin and stability range with increasing respective parameters will be plotted and then, the last section discusses the flow-phenomenon after probe insertion that is responsible for degradation in performance.

### 3.1 Clean flow validation

The current study utilizes a transonic speed rotor, namely Rotor-37, having a peak efficiency and total pressure ratio of 0.876 & 2.106, respectively at 20.19*kg/sec* while rotating at

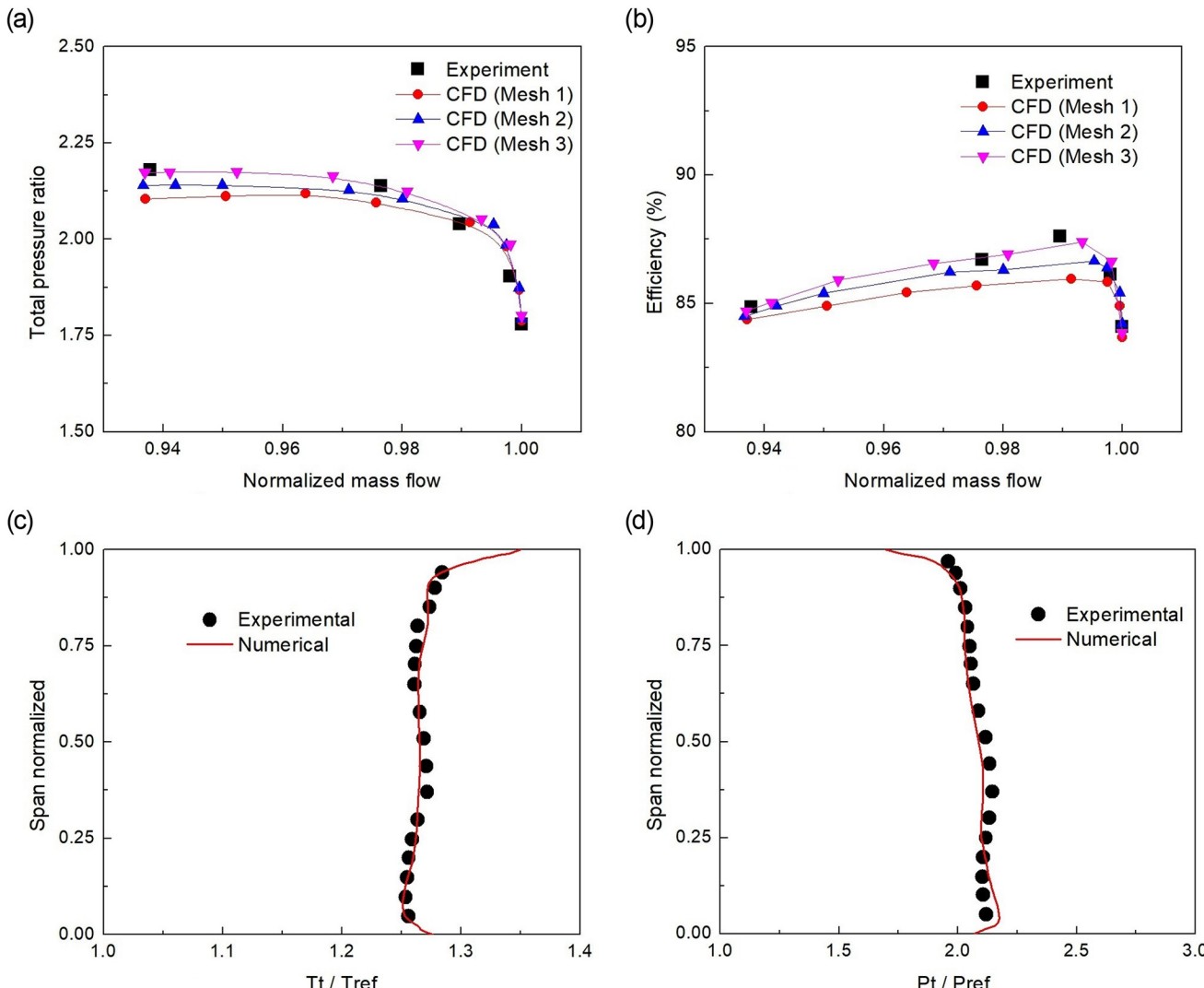

**Fig 2.** Clean flow performance characteristic curves of (a) Total Pressure ratio, (b)Adiabatic efficiency, and Radial distributions of (c) total temperature ratio, (d) total pressure ratio at rotor downstream with fine mesh.

17188.7$rev/min$. Suder et al. [26, 27] provided information about its experimental stations. Fig 1 shows the measurement stations and the CFD model with boundary conditions.

The clean flow case without the probe needs to be validated first before proceeding towards the analysis with the probe. So, the characteristics of total pressure ratio and adiabatic efficiency (Fig 2(a) and 2(b)) were generated with three different mesh sizes of 0.62, 1.2, and 1.9 million elements, respectively. The mesh size affected the performance prediction, particularly when approaching near the stall point. The fine mesh (Mesh 3) depicted a solution closer to the experimental data, and a nice validation was achieved. Moreover, (Fig 2(c) and 2(d)) shows the downstream rotor radial variations in terms of total temperature and total pressure respectively. Whereas for normalizing, the $m_{choke}$, $T_{ref}$ and $P_{ref}$ values were 20.93$kg/s$, 288$K$ and 101.325$kPa$, respectively. The results indicated a good comparison with the available experimental data [20, 29]. All these validation results of the unobstructed flow case form a

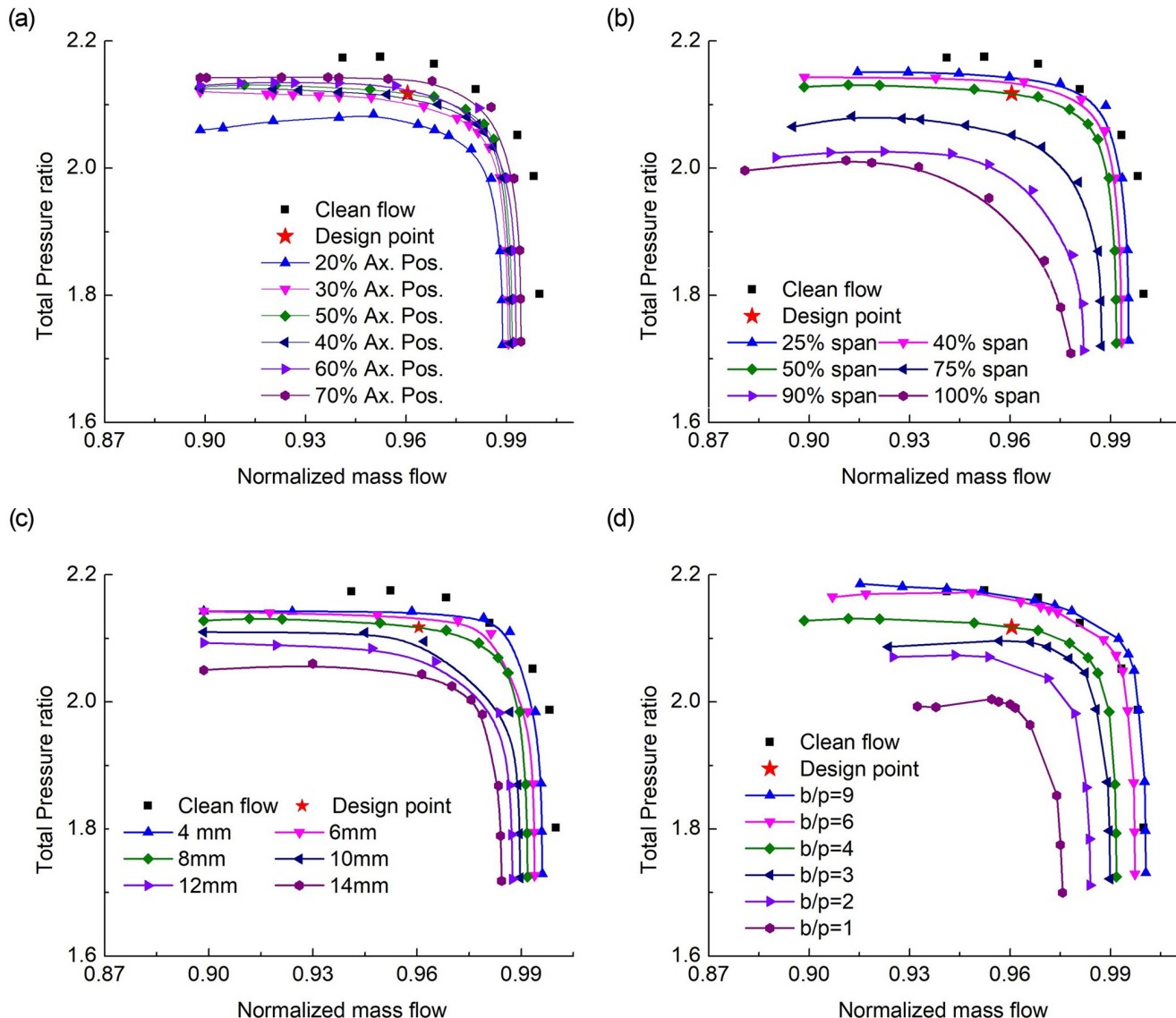

**Fig 3.** Total pressure ratio characteristics for effects of (a) streamwise axial probe positions, (b) Spanwise probe immersion, (c) probe sizes, and (d) total number of probes in full annulus.

benchmark for further research in the presence of probe and, subsequently, its influence on compressor performance.

## 3.2 Rotor performance with the probe

The probe is now introduced in the rotor upstream and the performance characteristics are generated in terms of total pressure ratio (Fig 3), and adiabatic efficiency (Fig 4). The major points are as under:

- Placing the probe at different axial positions in the rotor upstream shows different characteristics. The closest position (*i.e.*, 20% axial position) depicted a significant drop in total pressure ratio and efficiency. The significant deflection appears while approaching the stall

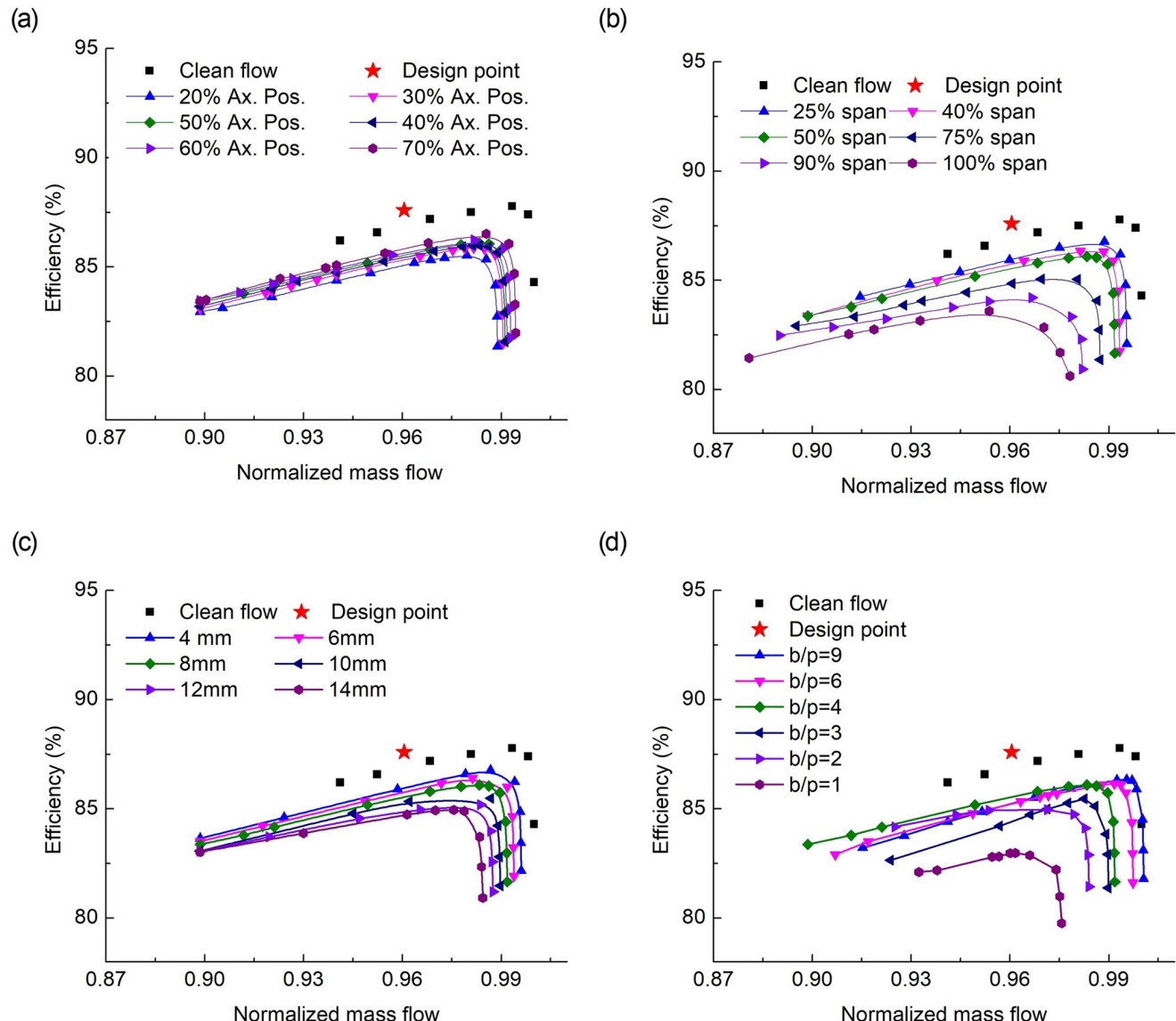

**Fig 4.** Adiabatic efficiency characteristics for effects of (a) streamwise axial probe positions, (b) Spanwise probe immersion, (c) probe sizes, and (d) total number of probes in full annulus.

regime. Also, placing the probe at an axial distance of about half blade chord or higher, makes a lesser wake effect and the characteristics map gets closer to the clean flow scenario.

- The radial immersion depths indicate that the total pressure ratio of the under-discussion rotor can be achieved at half span intrusion while the probe is located half blade chord upstream. It further shows that when the probe is inserted at a 25% span, the drop is minimum compared to the clean flow case. But as the immersion depth increases the drop maximizes along with the considerable drop in choking mass flow.

- The probe size plays a vital role in the flow distortions downstream, as can be seen from Figs 3c and 4c. The minimum the size, the closer it is to the clean flow characteristics. It means

that the flow blockage (Fig 7b) enhances with the increase of probe size that increases the Mach number and becomes a source for a decrease in the total pressure ratio and efficiency.

- The probe number in the full annulus also varies the performance. As increasing the total-blades to total-probes ratio means a reduction in the area blockage offered to the incoming flow. Although the adiabatic efficiency drop in all the cases involving probe but it significantly drops in increasing probe number and increasing probe-radial depth. The maps also show that besides degrading performance, the design total pressure ratio can be achieved with a total of 9 probes of 8mm size (Fig 3c and 3d).

### 3.3 Stability range and stall margin

The stall margin [30] and stability range [31] were calculated using the Eqs 6 and 7 respectively. The stall margin is dependent on peak efficiency and stall point data. Where, Pr and $\dot{m}$ represents the total pressure ratio and mass flow rate, respectively. While the subscripts represents the data at peak efficiency and stall points.

$$S.M = \frac{Pr_{stall} * \dot{m}_{peak}}{\dot{m}_{stall} * Pr_{peak}} - 1 \tag{6}$$

$$S.R = \frac{\dot{m}_{chock} - \dot{m}_{stall}}{\dot{m}_{chock}} \tag{7}$$

Fig 5 shows the variations for all the cases under discussion. The stall margin improves with locating the probe away from the rotor leading edge. The more the radial immersion depth, the lesser the stall margin but improved stability range. Besides, increasing the probe size decreases the stability with an increase in stall margin from 6*mm* to 10*mm*. However, increasing the total number of probes decrease both the compressor stall margin and its stability. It was therefore depicted that lesser total probes should be deployed for measurements to have a better compressor performance.

### 3.4 Flow phenomenon after probe insertion

This section discusses the changes in the total pressure ratio at peak efficiency, the peak efficiency, choking mass flow, and efficiency at clean flow stalling mass flow rate; due to the insertion of the probe in the rotor-upstream. The probe parameters were varied to analyze the effects on performance. Fig 6 represents a comprehensive analysis of all the parameters versus the corresponding increase in the parameter. The abscissa is represented on a normalized scale of 0–1. All the parameters (probe size, radial depth and axial distance increase) were normalized with the blade span height, while the parameter increase for the total number of probes represents the total-probes to total-blades ratio.

The increase in axial gap between the probe and blade leading edge, shows an increase in the peak efficiency along with the total pressure ratio at this condition. Also, the drop in choke mass flow minimizes with an increase in the upstream axial probe location, which means a reduction in flow blocking effect. The overall relative increase in radial depths in the flow-field, also drops the total pressure ratio at their respective peak efficiencies. However, increasing the radial probe immersion beyond half the blade span, decreases the total pressure ratio significantly.

The presence of the probe causes the adiabatic efficiency to drop in all the cases relative to the clean flow case. However, at the near stall mass flow rate of clean flow case, all with the

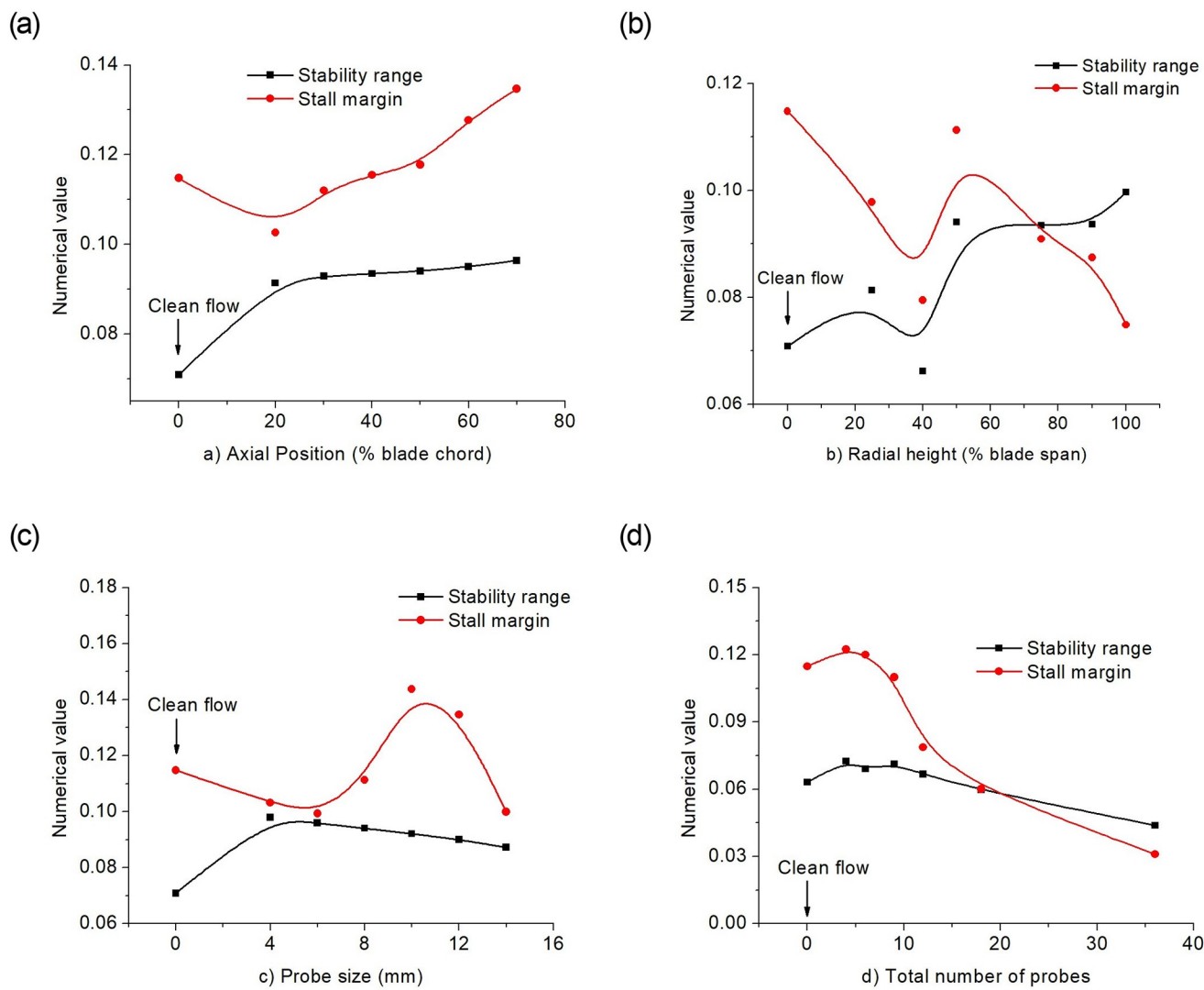

**Fig 5.** Stall margin and stability range for (a) streamwise axial probe positions, (b) Spanwise probe immersion, (c) probe sizes, and (d) total number of probes in full annulus.

probe cases were normal working. This is due to the probe blockage that decreased the mass flow rate and shifted the characteristics map towards the left, along with their respective peak efficiency points (Fig 4).

Although the mass flow drop near choke, was observed for all the cases with the probe but reducing the probes to a total of 4 in the full annulus, shows a slight increase in the mass flow relative to the clean flow. Besides, the total pressure ratio at its peak efficiency was also higher, although the peak efficiency value was lower than the clean flow case.

Due to large simulated data points and in order to show the flow-phenomenon responsible for degrading performance, the flow-field is shown on selective parameter data to have a distinct view. It can be seen from Fig 7 that the airflow bypasses the bottom of the strut to form an obvious acceleration area, and the larger the diameter of the stem, the larger the acceleration area. The radial influence along the probe rod starts to increase in full immersion (Fig 7a), along with an increase of rotational energy behind it. During the propagation of the rod trail

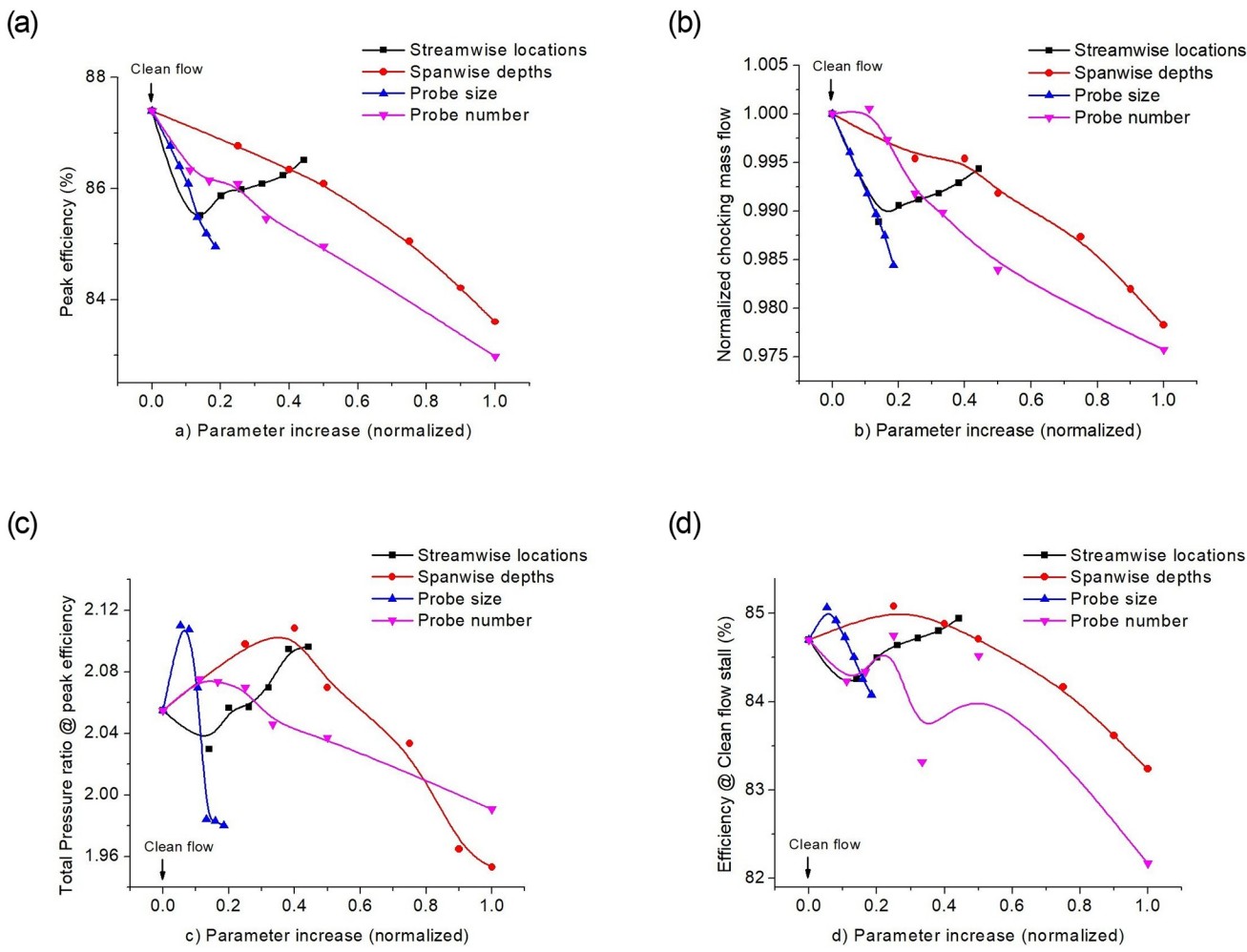

**Fig 6.** Variations in (a) peak adiabatic efficiency, (b) mass flow at the choke, (c) total pressure ratio at peak efficiency, (d) efficiency at stall mass flow rate of clean flow; versus the respective increase in parameter values.

wake in the downstream, the trail causes the vortical leakage flow to deflect in the channel behind the rod, and the larger the diameter of the rod, the more serious the leakage vortex deflection (Fig 7b). The deflection of the leakage vortex means that its influence range is expanded and the flow channel is more seriously blocked. Fig 7d and 7e shows the effects of four intrusive probe heights (25%, 50%, 75% & full span) relative to clean flow and indicates that the influence range in the circumferential direction has been expanded resulting in the production of tip leakage vortex at rotor trailing edge with increasing immersion heights. It further indicates that the trail of the stem has a specific pressure weakening wake effect in the downstream (Fig 7c), and the larger the diameter of the strut, the stronger the distortion effect that enhances the loss in the total pressure ratio of the compressor rotor.

## 4 Conclusions

The influences on transonic compressor performance have been studied through numerical CFD simulations. The clean flow data served as a reference to compare the effects of the probe presence. It was revealed that the upstream probe presence could influence the overall predicted compressor performance. The parameters like probe size, total probe number, axial

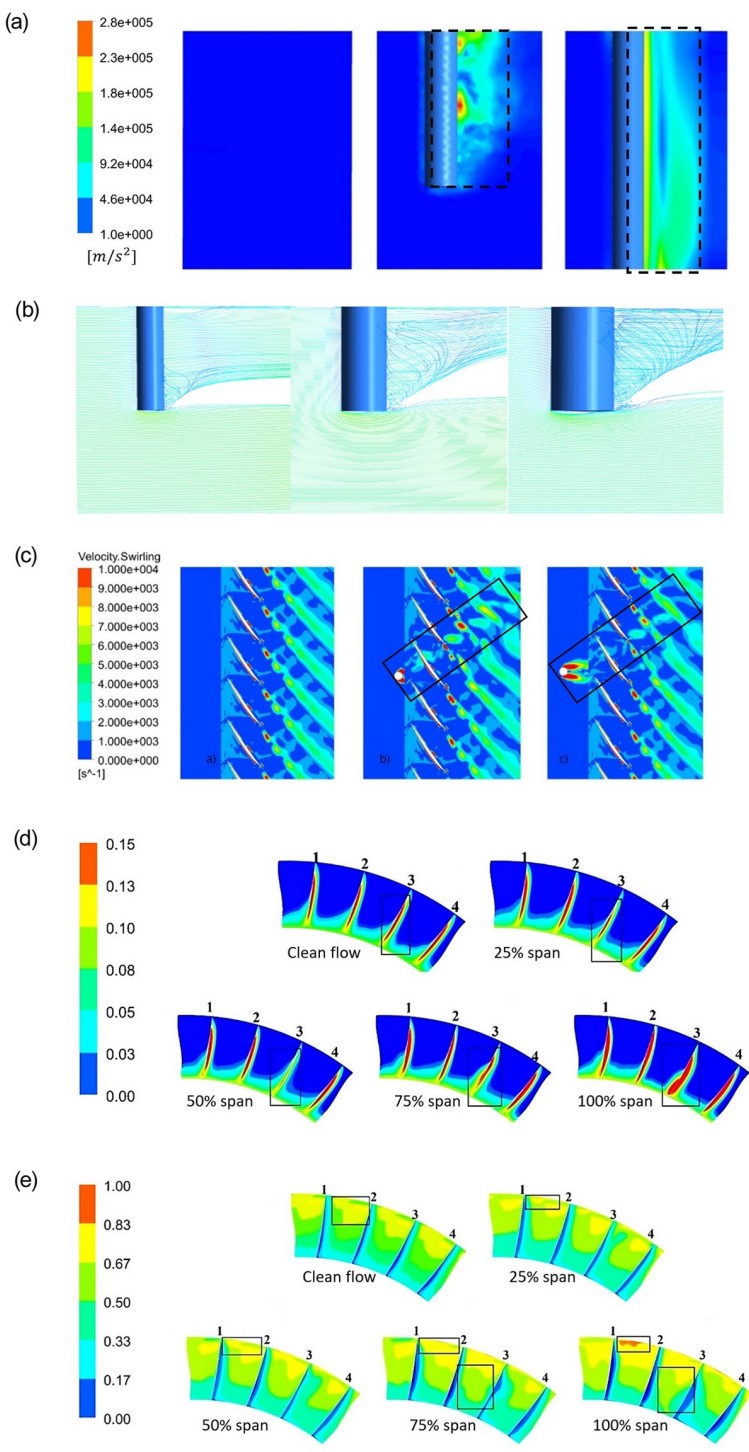

**Fig 7.** Flow phenomenon of clean flow, half and full probe immersion in terms of (a) Streamwise rothalpy gradient, (b) velocity streamlines with bigger probe sizes; (c) Velocity swirling strength for closest and far probe locations; Variations for clean flow and relative probe immersions; (d) Radial mach at rotor TE, and (e) Circumferential Mach distribution on rotor TE surface.

probe placement, and probe intrusion depth are often neglected while taking measurements. The improper selection of these parameters can impair the measurements and can degrade the rotor performance resulting in lower stall margin, ultimately affecting the end measurement result. The major points observed are as under:

1. The Mach number will increase as the probe is inserted into the flow-field. This increase further enhances with relative increasing probe-depths. In full immersion, the radial vortical flow starts to form behind the probe-rod in addition to axial that decreases the total pressure ratio drastically. The increase in the total number of probes also increases the frontal blockage area that reduced the performance characteristics both near stall and choke. So, in order to have a low wake and reduced blocking effect, the total number of probes deployed should be kept as low as possible.

2. The stall margin is dependent on the surge point and the peak efficiency data. The probe presence increases the blocking effect significantly with a larger mass flow rate (*i.e.*, near choke point), and shifts the characteristics curve towards left. It was due to the enhancement in the vortical influence regime in the downstream that increased the blockage.

3. The minimum probe size enhanced the stability range, but the stall margin with an 8*mm* probe size is closer to the clean flow case. The wake flow deflection enhanced with the greater probe size (Fig 7b) that degraded the rotor performance.

4. Increasing the axial distance between the upstream probe and blade leading edge, increase the stall margin, and the characteristics curve becomes closer to the clean flow data. It was further seen that in order to achieve the design total pressure ratio of the transonic compressor rotor under discussion, the probe should be placed at about half chord upstream while half blade span intruded with a total of 9 probes. While reducing probes to a total of 4, increases mass flow at choke slightly with a higher stall margin than clean flow.

5. The peak adiabatic efficiency, along with the total pressure ratio at this condition, drops with a relative increase in all the cases with the probe. It was further seen that besides lowering of stall margin, the mass flow rate at which the clean flow stalls is the normal working mass flow for the cases with the probe.

## Author Contributions

**Funding acquisition:** Hongwei Ma.

**Investigation:** Asad Islam.

**Methodology:** Asad Islam.

**Supervision:** Hongwei Ma.

**Validation:** Asad Islam.

**Writing – original draft:** Asad Islam.

**Writing – review & editing:** Asad Islam, Hongwei Ma.

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
