## [Decision Letter · Decision Letter 0]

23 Dec 2020

PONE-D-20-26128

Numerical study of Probe Parameters on Performance of a Transonic Axial Compressor

PLOS ONE

Dear Dr. Islam,

Thank you for submitting your manuscript to PLOS ONE. After careful consideration, we feel that it has merit but does not fully meet PLOS ONE’s publication criteria as it currently stands. Therefore, we invite you to submit a revised version of the manuscript that addresses the points raised during the review process.

We look forward to receiving your revised manuscript.

Kind regards,

Hongbing Ding, Ph.D.

Academic Editor

PLOS ONE

Additional Editor Comments:

Thank you for submitting your manuscript to PLOS ONE. The reviewers recommend reconsideration of your paper following minor revision. I invite you to resubmit your manuscript after addressing all reviewer comments.

Additional comments from staff editors:

Please note that adding all the additional references as suggested by the reviewer is not necessary for further consideration. We would recommend that you ensure that your reference list is balanced and relevant to your study.

Reviewers' comments:

Reviewer's Responses to Questions

**Comments to the Author**

1. Is the manuscript technically sound, and do the data support the conclusions?

Reviewer #1: Yes

Reviewer #2: Yes

2. Has the statistical analysis been performed appropriately and rigorously? 

Reviewer #1: Yes

Reviewer #2: Yes

3. Have the authors made all data underlying the findings in their manuscript fully available?

Reviewer #1: Yes

Reviewer #2: Yes

4. Is the manuscript presented in an intelligible fashion and written in standard English?

Reviewer #1: Yes

Reviewer #2: Yes

5. Review Comments to the Author

Reviewer #1: In this manuscript, the effect of the probe parameter on compressor performance is investigated, which is meaningful for compressor. The English looks sound and be well written. The results is meaningful and correct. But the reviewer has some suggestions about this paper.

1. how many is the reference pressure in numerical process?

2. the keywords need be re-checked carefully. ANSYS-CFX should be deleted.

3. in Introduction, the author should add some references about the non-equilibrium condensation in compressor, such as [DOI: 10.1016/j.applthermaleng.2020.115090], [DOI: 10.1016/j.ijmultiphaseflow.2019.03.005], and so on.

4. in computational section, the control equations should be presented.

5. in Figure 1, the figure1(d) need be re-charted to increase resolution.

6. the resolution of Figure 7 is too low. you can refer the reference [DOI: 10.1016/j.energy.2019.115982], [DOI: 10.1016/j.ijmultiphaseflow.2019.103083].

7. the figure 6 and 7 should be re-charted, referring the reference [DOI: 10.1016/j.applthermaleng.2019.114388], [DOI: 10.1016/j.enconman.2018.12.001].

Reviewer #2: Probes have been an essential part of compressor testing, but their presence also affects the flow-structure and the compressor performance as well. This paper has tried to well address the novel aspects associated with the probe using numerical simulations. The work is very interesting and valuable. The reviewer has some minor suggestions below:

1. The resolution for Figures (1d, 5 &6) should be enhanced to at least 300dpi.

2. The authors should mention the “Pref” and “Tref” values in text, which they used to plot Fig.1(c,d), and also the mass flow rate value used for normalizing.

3. The Figures 3&4 which show the characteristics maps for total pressure ratio and efficiency is currently a continuous line which is right but, it should display like “respective points connected with lines” (from choke to near stall).

6. PLOS authors have the option to publish the peer review history of their article (what does this mean?). If published, this will include your full peer review and any attached files.

Reviewer #1: No

Reviewer #2: No

---

## [Author Response · Author response to Decision Letter 0]

30 Dec 2020

Dear Editor,

We want to submit the Revision of our manuscript entitled “Numerical Study of Probe Parameters on Performance of a Transonic Axial Compressor”. 

The authors would like to thank the respected editors and reviewers for the comments & suggestions for improving our manuscript. Following are the revisions made for your kind considerations:

1. The authors have studied all the comments carefully and have made corrections accordingly. Revised portions are highlighted. (i.e., Revised Manuscript with track changes.pdf). And the responses to each of the reviewer comments are addressed in the rebuttal document (i.e., Response to reviewers.pdf).

2. For the “Data Availability Statement: All relevant data are within the manuscript”.

3. The paper is now formatted according to the PLOS journal formatting style and uploaded as PDF in the online submission system (i.e., Manuscript.pdf).

Hope to hear from you soon. 

Best Regards, 

Asad Islam

---

## [Editor Report · Decision Letter 1]

7 Jan 2021

Numerical Study of Probe Parameters on Performance of a Transonic Axial Compressor

PONE-D-20-26128R1

Dear Dr. Islam,

We’re pleased to inform you that your manuscript has been judged scientifically suitable for publication and will be formally accepted for publication once it meets all outstanding technical requirements.

Kind regards,

Hongbing Ding, Ph.D.

Academic Editor

PLOS ONE

Additional Editor Comments (optional):

The authors have done a good job in revising the manuscript. Now it can be accepted for publication in PLOS ONE.
---

## [Editor Report · Acceptance letter]

14 Jan 2021

PONE-D-20-26128R1 

Numerical Study of Probe Parameters on Performance of a Transonic Axial Compressor 

Dear Dr. Islam:

I'm pleased to inform you that your manuscript has been deemed suitable for publication in PLOS ONE. Congratulations! Your manuscript is now with our production department. 

Kind regards, 

on behalf of

Professor Hongbing Ding 

Academic Editor

PLOS ONE